# Does practice analysis agree with the ambulatory care sensitive conditions' list of avoidable unplanned admissions?: a cross-sectional study in the East of England

Robert Fleetcroft,[1] Antonia Hardcastle,[2] Nicholas Steel,[3] Gill M Price,[3] Sarah Purdy,[4] Alistair Lipp,[5] Phyo Kyaw Myint,[6] Amanda Howe[3]

[1]The Surgery, Acle Medical Partnership, Acle, UK
[2]Research and Development Department, The Queen Elizabeth Hospital King's Lynn NHS Foundation Trust, King's Lynn, UK
[3]Norwich Medical School, University of East Anglia, Norwich, UK
[4]School of Social and Community Medicine, University of Bristol, Bristol, UK
[5]NHS England Midlands & East [East], Victoria House, Capital Park, Fulbourn, UK
[6]School of Medicine, Medical Sciences & Nutrition, University of Aberdeen, Aberdeen, UK

**Correspondence to**
Dr Robert Fleetcroft;
robert.fleetcroft@nhs.net

## ABSTRACT

**Objectives** To use significant event audits (SEAs) in primary care to determine which of a sample of emergency (unplanned) admissions were potentially avoidable; and compare with the National Health Service (NHS) list of ambulatory care sensitive conditions (ACSCs).

**Design** Analysis of unplanned medical admissions randomly identified in secondary care.

**Setting** Primary care in the East of England.

**Participants** 20 general practice teams trained to use SEA on unplanned admissions to identify potentially preventable factors.

**Interventions** SEA of admissions.

**Main outcome measures** Level of agreement between those admissions identified as potentially preventable by SEA and the NHS ACSC list.

**Results** 132 (26%) of randomly selected patients with unplanned admissions gave consent and an SEA was performed by their primary practice team. 130 SEA reports had sufficient data for our analysis. Practices concluded that 17 (13%) admissions were potentially preventable. The NHS ACSC list identified 36 admissions (28%) as potentially preventable. There was a low level of agreement between the practices and the NHS list as to which admissions were preventable (kappa=0.253). The ACSC list consisted mainly of respiratory admissions whereas the practice list identified a wider range of cases and identified context-specific factors as important.

**Conclusions** There was disagreement between the NHS list and practice conclusions of potentially avoidable admissions. The SEAs suggest that the pathway into unplanned admission may be less dependent on the condition than on context-specific factors, and the assumption that unplanned admissions for ACSCs are reasonable indicators of performance for primary care may not be valid.

## INTRODUCTION

'Emergency' admissions to hospital are unplanned—unlike elective admissions, which are planned for specific times and dates. There were 5.3 million unplanned admissions to English hospitals in 2013 representing 67% of hospital bed days and costing £12.5 billion.[1] In England, unplanned admissions have continued to rise by approximately 2% each year over the past decade.[2 3] Unplanned admissions create difficulties for those responsible for planning and delivering services, and they are distressing for patients and their families.[4]

It is claimed that many medical 'unplanned' admissions could be prevented by being managed differently in primary care. Others have distinguished between avoidability (through having alternatives to admission at the time of event, eg, intermediate care) and preventability (through preventive actions usually in primary care, eg, maximising asthma control).[5] The latter are called 'ambulatory care sensitive conditions' (ACSCs).[6] National Health Service (NHS) England has defined ACSCs as conditions where effective community care and case management can help prevent the need for hospital admission,[7] and these are currently used as a performance indicator for primary care.[8 9]

**Table 1** List of ACSCs and ICD-10 codes[8]

| ACSC group name | ICD-10 codes |
| --- | --- |
| Influenza and pneumonia | J10, J11, J13, J14, J15.3, J15.4, J15.7, J15.9, J16.8, J18.1, J18.8 |
| Other vaccine preventable | A35–A37, A80, B05, B06, B16.1, B16.9, B18.0, B18.1, B26, G00.0, M01.4 |
| Asthma | J45, J46 |
| Congestive heart failure | I11.0, I50, J81 |
| Diabetes complications | E10.0–E10.8, E11.0–E11.8, E12.0–E12.8, E13.0–E13.8, E14.0–E14.8 |
| Chronic obstructive pulmonary disease | J20, J41–J44, J47 |
| Angina | I20, I24.0, I24.8, I24.9 |
| Iron deficiency anaemia | D50.1, D50.8, D50.9 |
| Hypertension | I10, I11.9 |
| Nutritional deficiencies | E40–E43, E55.0, E64.3 |
| Dehydration and gastroenteritis | E86, K52.2, K52.8, K52.9 |
| Pyelonephritis | N10–N12, N13.6 |
| Perforated/bleeding ulcer | K25.0–K25.2, K25.4–K25.6, K26.0–K26.2, K26.4–K26.6, K27.0–K27.2, K27.4–K27.6, K28.0–K28.2, K28.4–K28.6 |
| Cellulitis | L03, L04, L08.0, L08.8, L08.9, L88, L98.0 |
| Pelvic inflammatory disease | N70, N73, N74 |
| Ear, nose and throat infections | H66, H67, J02, J03, J06, J31.2 |
| Dental conditions | A69.0, K02–K06, K08, K09.8, K09.9, K12, K13 |
| Convulsions and epilepsy | G40, G41, R56, O15 |
| Gangrene | R02 |

ACSC, ambulatory care sensitive condition; ICD-10, International Classification of Diseases, 10th Revision.

In an attempt to reduce unplanned admissions, lists of ACSCs have been produced by many countries including the USA, Spain, the UK, Australia and New Zealand.[10–14] The methodology used to identify lists of ACSCs typically includes consensus opinions of expert physician panels drawn from secondary and primary care, sometimes in conjunction with literature searches for guidelines in the best practice.[6 15 16]

The current UK NHS list contains 19 conditions (focusing largely on medical rather than surgical conditions) where unplanned admission is thought to be potentially avoidable (table 1).[9] This NHS list of ACSCs represents about 20% of all unplanned admissions in the UK,[17] and is similar to other international lists of ACSCs, which constitute between 10% and 20% of all unplanned admissions.[18] The NHS list was developed from previous lists of ACSCs in use in the USA[19]; it includes some conditions identified in a study based in England which used three expert panels. These panels included primary care and hospital clinicians, who reviewed a list of 174 disease codes (International Classification of Diseases, Ninth Revision) of clinical conditions related to unplanned admission.[12] A consensus was then reached on those conditions where unplanned admission might have been prevented. Conditions were defined as a 'definite ACSC' if 70% or more of admissions were judged preventable by better prevention or by better management of that condition. However, the panels looked only at conditions, not patient-specific case details.

The extent to which unplanned admissions which have been identified as an ACSC can actually be prevented in practice is not known.[20] A systematic review in 2000 reported a wide variation of estimates of the rates of unplanned admissions which were thought to be preventable, ranging from <1% to 29% for unplanned admissions to medical wards in the UK.[20]

Overall, unplanned admission rates for ACSCs have been rising, increasing in England by 40% over the 10 years from 2001 to 2011, but not for all conditions; for example, rates of admissions for angina and congestive heart failure have fallen by 33%, though rates for chronic obstructive pulmonary disease (COPD) have risen by 25% over this period.[17]

This literature, therefore, suggests that the ACSC approach may be oversimplistic. Another set of factors thought to contribute to unplanned admission is overall patient vulnerability, which may include both physical and mental comorbidities.[21] Indeed, many risk prediction models are being developed to allow early interventions in community settings, and all of these use more factors than ACSC alone.[22] There is another discourse about the availability of adequate services to support patients outside hospital in a period of deterioration where unplanned admissions may occur if additional local support cannot be put in place.[23]

Many studies use patient record review to look at pathways to admission that are potentially avoidable. Analysis of clinical records is already widely used in healthcare, examples being significant event audit (SEA) in primary care and root cause analysis in secondary care to examine unexpected outcomes and in identifying avoidable deaths.[24–26] Research using hospital physicians and senior administrators performing detailed notes reviews of patients who have experienced an unplanned admission (rather than just diagnostic codes) has reported lower rates of preventability of admissions than the ACSC lists, at between 6% and 10%.[27–29] However, most of this research is based in secondary care, and there is little evidence of the primary care perspective on preventable admissions.[20]

We, therefore, aimed to fill this gap in the research by using a detailed notes review (SEA) by the primary care

health team who had looked after that patient to identify potentially preventable factors in unplanned admissions; and then comparing the estimated avoidability of a list of unplanned admissions using two methods of assessment, those admissions deemed avoidable by the ACSC list and by a detailed notes review.

## METHODS

This research was part of a National Institute for Health Research study titled 'Can a practice based approach using Significant Event Audit identify key factors that might reduce avoidable non-elective hospital admissions?'. One hundred and ten practices were approached in the Norfolk and Waveney area of East Anglia in the UK, and 20 practices were recruited. The practices included in the study were representative of English practices with respect to practice size and markers of deprivation, however, they were more likely to be rural. The practices included had a mean of 8640 registered adult patients (range 3362–16 148, national mean 8316). Seventy per cent practices were classified 'urban', 20% practices 'town and fringe' and 10% were 'rural' (national averages 85%, 11% and 3%, respectively). The mean Index of Multiple Deprivation score for practices was 21.1 (range 7.3–51.9, national mean 24.0, higher is more deprived).

These practices used one of three local hospitals for unplanned admissions, these being the James Paget University Hospitals NHS Foundation Trust, the Norfolk and Norwich University Hospitals NHS Foundation Trust and the Queen Elizabeth Hospital King's Lynn NHS Foundation Trust. Practices were recruited who were willing to discuss unplanned admissions from their own practice and carry out a type of case notes review called Significant event audit (SEA) on a subgroup of these patients after discharge. SEA entails a detailed review of individual case notes of patients by their practice team. We chose this approach because SEA is a technique which is already commonly used to analyse patient safety incidents and unexpected outcomes both by general practice for appraisal, although it is not routinely used to analyse unplanned admissions. SEA entails individual episodes being analysed in a systematic and detailed way by the practice team to ascertain what can be learnt about the overall quality of care, and to indicate any changes that might lead to future improvements.[30]

Each practice identified one lead clinician (usually a general practitioner (GP)) and one designated administrator who were responsible for running and facilitating SEA meetings within that practice, and writing SEA reports. The lead clinician and administrator attended one of two training sessions led by an experienced GP educator, where training was given both in their roles as practice facilitators of SEA and in writing a summary report for each case. This training was based on the guidance issued by the National Patient Safety Agency for the conduct of SEA, and this guidance is recommended by the Royal College of General Practitioners as best practice.[30]

Data on all unplanned admissions to medical wards were collected by a data clerk at the three participating hospitals; from these admissions, one was randomly identified for each practice in each week. The practice then approached that patient for consent to participate in the study.

Cases were discussed at the practice clinical meeting in primary care facilitated by the trained GP and administrator. A summary of each case was written up by the administrator with the guidance of the lead GP based on the salient points that have arisen from the practice clinical meeting. The report included information on how the immediate clinical decision to admit happened; whether the admission could have been avoided by factors under primary care control (yes or no); whether patient, practice or systems factors may have been contributory; what changes have been made to avoid this recurring; how long those changes would take to put in place and see their impact; in addition, how likely it was that similar circumstances could contribute to further admissions. Feedback was given to each practice on the quality of the SEA report after they had performed the first three SEAs.

The ICD-10 disease code for the primary cause of admission was identified for each unplanned admission determined at discharge from the hospital database so that the cases could be categorised into those with an ACSC and those without an ACSC.[9] Preventability of each admission by the practice teams was based on their response to the question in the SEA form 'Whether the admission could have been avoided by factors under primary care control.'. An inter-rater reliability analysis using the Cohen's kappa statistic was performed to determine the consistency between admissions considered being preventable using the ACSC list and those considered to be preventable by a practice team using SEA. Both of these measures of preventable admissions are independent. These data were computed in SPSS Statistics V.22.

### Patient involvement

We involved patients and public through the PPIRes group (public and patient involvement in research) in the design, application process and conduct of the research project that analysed the unplanned admissions. Earlier versions of this paper was discussed by the steering group meetings which included patient representatives. Research outputs will be disseminated though local networks to patients.

## RESULTS

There were 3355 unplanned medical admissions to acute trusts from the participating 20 practices over the course of the study. Nineteen practices (95%) completed the study; one practice withdrew due to new ownership and staff shortages. The practices had a mean of 8640 patients each (range 3362–16 148, SD 3864, national mean 8316), and a median Index of Multiple Deprivation 2010 of 19.72 (range 7.32–52.0, national mean 24, higher

number indicates more deprived population). Data on rurality of practices reported that 14 practices were classified as 'urban', 4 practices were 'town and fringe' and 2 practices were 'village'. Out of the 507 patients randomly selected, 132 (26%) gave their consent and an SEA was performed and a report written. In these SEA reports, data on preventability were missing for two patients, leaving 130 reports for analysis (98%).

Table 2 displays the list of ICD-10 disease codes for each of 130 admissions in the study, whether the condition was included in the ACSC list, and whether the SEAs carried out by practices concluded that these admissions were potentially avoidable. Of these 130 admissions, the practices concluded that 17 (13%) were potentially preventable by factors under primary care control. Using the NHS ACSC list of avoidable conditions, 36 admissions (28%) were categorised to be potentially preventable. The inter-rater agreement coefficient, kappa, between avoidability conclusion by practices using SEA and preventability according to the NHS ACSC code was 0.253, where a value of 1 represents perfect agreement, and a value of 0 represents no agreement 'above that which might be expected by chance'. Guides to interpretation of values of kappa suggest only 'fair' agreement between raters for 0.253, and we would expect a moderate or substantial agreement with a kappa between 0.41 and 0.80.[31–33] Put another way, in this series there was agreement in 75% of cases, which was only 8% above the expected agreement by chance of 67%. Those admissions identified using the NHS ACSC list as potentially preventable mainly consisted of respiratory admissions, these being pneumonia 14 cases (39%), COPD 5 cases (14%), asthma 3 cases (8%) and bronchiectasis 1 case (3%). Practice teams identified fewer respiratory causes as potentially preventable—pneumonia four cases (24%), asthma one case (6%), COPD one case (6%) and bronchiectasis one case (6%). Practices identified a wider range of cases as potentially preventable including admission with codes for constipation, anxiety and cancer.

Table 3 lists the cases where there was disagreement between the ACSC list and SEA as to whether the admission could have been prevented. There were seven cases where SEA determined admission was avoidable, but the key causal factors identified were not on the ACSC list of preventable admissions. In these cases, SEA identified that patient factors were present in 6/7, practitioner factors were present in 3/7 cases and systems factors were present in 4/7 cases. Free text comments also showed patient factors, such as presenting at A&E rather than contacting their own GP; failure to seek timely advice from their GP in a case of sepsis and increasing shortness of breath and failing to attend routine diabetes check-up with their general practice, leading to a loss of control of diabetes. Comments for practitioner factors included, for example, failure to prescribe a PPI to a patient on aspirin, and admission by the Out of Hours (OOHs) service where the GP felt home management was feasible. Comments for systems factors included lack of communication between

the practice and OOHs service in two cases. Five of these seven cases occurred outside normal working hours using the 'out-of-hours' services through 111.

Of the total 36 cases in our study in which the ACSC list identified an admission as potentially avoidable, there were 25 (69%) cases in which the SEA did not. These included 5 cases of heart failure, 11 cases of lobar pneumonia, 2 cases of emphysema, 4 cases of COPD, 2 cases of asthma and 1 case of cellulitis. In these cases, SEA identified that patient factors leading to admission were present in 13/25, practitioner factors were present in 2/25 cases and systems factors were present in 6/25 cases. Despite these factors being present, the judgement was that admission was not preventable.

## DISCUSSION
### Statement of principal findings
There was disagreement on which admissions might be avoidable when comparing the NHS ACSC list with the list generated by the practice teams using SEA. These findings suggest that the avoidability of unplanned admission may be more dependent on the context-specific factors than the condition. These findings also suggest that in some cases the context may be at least as important as or more important than the diagnosis. The current use of a diagnostic label to identify potentially avoidable admissions (such as the ACSC list) might be problematic as a diagnostic label does not allow for different levels of severity of the condition, influential comorbidities such as dementia, and at which point in time the admission may have been avoidable—the condition may have deteriorated beyond prevention of admission if left 'too late'. This may suggest that the assumption that unplanned admissions for ACSCs are reasonable indicators of performance for primary care may not be valid—for example, this indicator may reflect the availability and quality of social care rather than the quality of primary care.[34] Our findings that many ACSC admissions may not be avoidable may also explain why ACSC-related unplanned admissions have continued to rise over the past years despite many efforts aimed to reduce these admissions.[35]

### Strengths and weaknesses of the study
The strength of the study was the analysis of the review of the patients record rather than administrative and ICD codes, and this analysis was performed soon after discharge by the primary care team looking after the patient, and as far as we are aware this is the first time this approach has been used in primary care. The unplanned admissions were randomly selected, and they were assessed by the practice team who looked after the patient at the time of admission. The proportion of admissions which were on the ACSC list (28%) is within the range identified in other studies and datasets[19 20] and may be higher than most estimates due to our study addressing medical admissions only. Training had been given to a lead clinician and administrator from each practice to

**Table 2** Emergency admission ICD-10 codes for each of 130 patients included in this study, and whether their admission was thought to be potentially avoidable by two methods

| ICD-10 disease codes | Description of disease codes | Determined by SEA to be potentially avoidable? | Included in the ACSC list as potentially avoidable? |
|---|---|---|---|
| A08.3 | Other viral enteritis | No | No |
| A09.0 | Other and unspecified infectious gastroenteritis and colitis, of infectious origin | No | No |
| A09.9 | Infectious gastroenteritis and colitis, unspecified | No | No |
| A41.8 | Other specified sepsis | No | No |
| B35.3 | Tinea pedis | No | No |
| C34.1 | Malignant neoplasm of upper lobe, bronchus or lung | No | No |
| C34.1 | Malignant neoplasm of upper lobe, bronchus or lung | Yes | No |
| C34.3 | Malignant neoplasm; lower lobe, bronchus or lung | No | No |
| C34.3 | Malignant neoplasm; lower lobe, bronchus or lung | No | No |
| C34.9 | Malignant neoplasm of bronchus and lung—unspecified | No | No |
| C78.0 | Secondary malignant neoplasm of lung | No | No |
| C79.5 | Secondary malignant neoplasm of bone and bone marrow | No | No |
| C91.1 | Chronic lymphocytic leukaemia of B-cell type | No | No |
| C91.1 | Chronic lymphocytic leukaemia of B-cell type | No | No |
| C92.0 | Acute myeloblastic leukaemia | No | No |
| C97 | Malignant neoplasms of independent (primary) multiple sites | No | No |
| D32.9 | Benign neoplasm: meninges, unspecified | No | No |
| D64.9 | Anaemia, unspecified | No | No |
| E10.9 | Type 1 diabetes mellitus without complications | Yes | No |
| F41.9 | Anxiety disorder, unspecified | Yes | No |
| G20 | Parkinson's disease | No | No |
| G20 | Parkinson's disease | No | No |
| G43.9 | Migraine—migraine, unspecified | No | No |
| H81.3 | Other peripheral vertigo | No | No |
| I21.0 | Acute transmural myocardial infarction of anterior wall | No | No |
| I21.0 | Acute transmural myocardial infarction of anterior wall | No | No |
| I21.0 | Acute transmural myocardial infarction of anterior wall | No | No |
| I21.1 | Acute transmural myocardial infarction of inferior wall | No | No |
| I21.1 | Acute transmural myocardial infarction of inferior wall | No | No |
| I21.1 | Acute transmural myocardial infarction of inferior wall | No | No |
| I21.4 | Acute subendocardial myocardial infarction | No | No |
| I21.9 | Acute myocardial infarction, unspecified | No | No |
| I21.9 | Acute myocardial infarction, unspecified | No | No |
| I21.9 | Acute myocardial infarction, unspecified | No | No |
| I21.9 | Acute myocardial infarction, unspecified | No | No |
| I21.9 | Acute myocardial infarction, unspecified | No | No |
| I21.9 | Acute myocardial infarction, unspecified | No | No |
| I21.9 | Acute myocardial infarction, unspecified | No | No |
| I26.9 | Pulmonary embolism without mention of acute cor pulmonale | No | No |
| I35.0 | Aortic (valve) stenosis | No | No |
| I44.2 | Atrioventricular block, complete | No | No |
| I46.0 | Cardiac arrest with successful resuscitation | No | No |

**Table 2** Continued

| ICD-10 disease codes | Description of disease codes | Determined by SEA to be potentially avoidable? | Included in the ACSC list as potentially avoidable? |
|---|---|---|---|
| I46.9 | Cardiac arrest, unspecified | No | No |
| I48 | Atrial fibrillation and flutter | No | No |
| I48 | Atrial fibrillation and flutter | No | No |
| I48 | Atrial fibrillation and flutter | No | No |
| I48 | Atrial fibrillation and flutter | No | No |
| I50.0 | Congestive heart failure | No | Yes |
| I50.0 | Congestive heart failure | No | Yes |
| I50.0 | Congestive heart failure | No | Yes |
| I50.0 | Congestive heart failure | No | Yes |
| I50.0 | Congestive heart failure | No | Yes |
| I50.0 | Congestive heart failure | Yes | Yes |
| I50.0 | Congestive heart failure | Yes | Yes |
| I60.2 | Subarachnoid haemorrhage from anterior communicating artery | No | No |
| I61.1 | Intracerebral haemorrhage in hemisphere, cortical | Yes | No |
| I61.9 | Intracerebral haemorrhage, unspecified | No | No |
| I63.5 | Cerebral infarction due to unspecified occlusion or stenosis of cerebral arteries | No | No |
| I63.9 | Cerebral infarction, unspecified | No | No |
| I63.9 | Cerebral infarction, unspecified | No | No |
| I63.9 | Cerebral infarction, unspecified | No | No |
| I64 | Stroke, not specified as haemorrhage or infarction | No | No |
| I95.1 | Orthostatic hypotension | No | No |
| J18.0 | Bronchopneumonia, unspecified | Yes | No |
| J18.0 | Bronchopneumonia, unspecified | No | No |
| J18.0 | Bronchopneumonia, unspecified | Data missing | No |
| J18.1 | Lobar pneumonia, unspecified | Yes | Yes |
| J18.1 | Lobar pneumonia, unspecified | No | Yes |
| J18.1 | Lobar pneumonia, unspecified | No | Yes |
| J18.1 | Lobar pneumonia, unspecified | No | Yes |
| J18.1 | Lobar pneumonia, unspecified | No | Yes |
| J18.1 | Lobar pneumonia, unspecified | No | Yes |
| J18.1 | Lobar pneumonia, unspecified | No | Yes |
| J18.1 | Lobar pneumonia, unspecified | No | Yes |
| J18.1 | Lobar pneumonia, unspecified | No | Yes |
| J18.1 | Lobar pneumonia, unspecified | No | Yes |
| J18.1 | Lobar pneumonia, unspecified | No | Yes |
| J18.1 | Lobar pneumonia, unspecified | Yes | Yes |
| J18.1 | Lobar pneumonia, unspecified | No | Yes |
| J18.1 | Lobar pneumonia, unspecified | Yes | Yes |
| J18.9 | Pneumonia, unspecified | No | No |
| J22 | Unspecified acute lower respiratory infection | No | No |
| J22 | Unspecified acute lower respiratory infection | No | No |
| J22 | Unspecified acute lower respiratory infection | No | No |

Continued

| ICD-10 disease codes | Description of disease codes | Determined by SEA to be potentially avoidable? | Included in the ACSC list as potentially avoidable? |
|---|---|---|---|
| J43.9 | Emphysema, unspecified | No | Yes |
| J43.9 | Emphysema, unspecified | No | Yes |
| J44.0 | Chronic obstruct pulmonary disease with acute lower respiratory infection | No | Yes |
| J44.0 | Chronic obstruct pulmonary disease with acute lower respiratory infection | No | Yes |
| J44.1 | Chronic obstruct pulmonary diseases with acute exacerbation, unspecified | No | Yes |
| J44.1 | Chronic obstruct pulmonary disease with acute exacerbation, unspecified | Yes | Yes |
| J44.1 | Chronic obstruct pulmonary disease with acute exacerbation, unspecified | No | Yes |
| J45.9 | Asthma, unspecified | Yes | Yes |
| J45.9 | Asthma, unspecified | No | Yes |
| J45.9 | Asthma, unspecified | No | Yes |
| J47 | Bronchiectasis | Yes | Yes |
| J69.0 | Pneumonitis due to food and vomit | No | No |
| J86.9 | Pyothorax—pyothorax without fistula | No | No |
| J93.8 | Pneumothorax—other pneumothorax | No | No |
| K22.2 | Oesophageal obstruction | No | No |
| K26.3 | Duodenal ulcer, acute without haemorrhage or perforation | Yes | No |
| K59.0 | Constipation | Yes | No |
| K70.3 | Alcoholic cirrhosis of liver | No | No |
| K70.3 | Alcoholic cirrhosis of liver | No | No |
| K80.1 | Calculus of gallbladder with other cholecystitis | No | No |
| K80.3 | Calculus of bile duct with cholangitis | No | No |
| K83.0 | Cholangitis | No | No |
| K92.2 | Gastrointestinal haemorrhage, unspecified | No | No |
| L03.1 | Cellulitis of other parts of limb | No | Yes |
| L03.1 | Cellulitis of other parts of limb | Yes | Yes |
| L03.1 | Cellulitis of other parts of limb | Data missing | Yes |
| L03.1 | Cellulitis of other parts of limb | Yes | Yes |
| L40.0 | Psoriasis vulgaris | No | No |
| N39.0 | Urinary tract infection, site not specified | No | No |
| N39.0 | Urinary tract infection, site not specified | No | No |
| R10.3 | Pain localised to other parts of lower abdomen | No | No |
| R10.3 | Pain localised to other parts of lower abdomen | No | No |
| R11 | Nausea and vomiting | No | No |
| R29.6 | Tendency to fall, not elsewhere classified | No | No |
| R33 | Retention of urine | No | No |
| R50.9 | Fever, unspecified | No | No |
| R51 | Headache | No | No |
| R55 | Syncope and collapse | No | No |
| R63.4 | Abnormal weight loss | No | No |
| S01.0 | Open wound of scalp | No | No |

**Table 2** Continued

**Table 2** Continued

| ICD-10 disease codes | Description of disease codes | Determined by SEA to be potentially avoidable? | Included in the ACSC list as potentially avoidable? |
|---|---|---|---|
| S01.8 | Open wound of other parts of head | No | No |
| S42.0 | Fracture of clavicle | No | No |
| S72.0 | Fracture of neck of femur | No | No |
| S72.0 | Fracture of neck of femur | No | No |
| S72.1 | Fracture of femur—pertrochanteric fracture | No | No |
| T39.1 | Poisoning by 4-aminophenol derivatives | No | No |
| T45.5 | Poisoning by anticoagulants | No | No |
| T84.0 | Mechanical complication of internal joint prosthesis | No | No |
| | Total numbers of admissions deemed avoidable | 17/132 (13%) | 36/132 (28%) |

ACSC, ambulatory care sensitive condition; ICD-10, International Classification of Diseases, 10th Revision; SEA, significant event audit.

analyse the cases in a standardised way, and feedback on the standard of SEA report was given to practices early in the study. The sample of practices was generally representative of the demography of English practices. Weaknesses include that the secondary care input to this process was limited to the discharge information received by primary care, and there was no input from the patient or where relevant the emergency services. This was a relatively small sample of unplanned admissions, and a larger sample will have provided more information on a greater range of conditions, on the types of preventable admission and what factors may be involved. Only 26% of randomly identified cases were analysed, this was mainly due to difficulties with gaining consent from patients, which

**Table 3** Cases where there were disagreement on preventability of admission, and whether patient practitioner or systems factors were identified in each case

| ICD-10 code for unplanned admission | Description of disease codes | Total number of cases | Case determined by SEA to be potentially avoidable? | Condition on the ACSC list as avoidable? | Number of cases where patient factors contributed to admission? | Number of cases where practitioner factors contributed to admission? | Number of cases where system factors contributed to admission? |
|---|---|---|---|---|---|---|---|
| I50.0 | Congestive heart failure | 5 | 0/5 | 5/5 | 1/5 | 0/5 | 1/5 |
| J18.0 | Bronchopneumonia, unspecified | 1 | Yes | No | Yes | No | No |
| J18.1 | Lobar pneumonia, unspecified | 11 | 0/11 | 11/11 | 5/11 | 0/11 | 1/11 |
| J43.9 | Emphysema, unspecified | 2 | 0/2 | 2/2 | 2/2 | 1/2 | 2/2 |
| J44.0 | Chronic obstruct pulmonary disease with acute lower respiratory infection | 2 | 0/2 | 2/2 | 2/2 | 0/2 | 0/2 |
| J44.1 | Chronic obstruct pulmonary diseases with acute exacerbation, unspecified | 2 | 0/2 | 2/2 | 2/2 | 0/2 | 2/2 |
| J45.9 | Asthma, unspecified | 2 | 0/2 | 2/2 | 1/2 | 1/2 | 0/2 |
| L03.1 | Cellulitis of other parts of limb | 1 | No | Yes | No | No | No |
| C34.1 | Malignant neoplasm of upper lobe, bronchus or lung | 1 | Yes | No | Yes | Yes | Yes |
| E10.9 | Type 1 diabetes mellitus without complications | 1 | Yes | No | Yes | No | Yes |
| F41.9 | Anxiety disorder, unspecified | 1 | Yes | No | Yes | Yes | Yes |
| I61.1 | Intracerebral haemorrhage in hemisphere, cortical | 1 | Yes | No | Yes | No | Yes |
| K26.3 | Duodenal ulcer, acute without haemorrhage or perforation | 1 | Yes | No | No | No | No |
| K59.0 | Constipation | 1 | Yes | No | Yes | Yes | No |

ACSC, ambulatory care sensitive condition; ICD-10, International Classification of Diseases, 10th Revision; SEA, significant event audit.

according to practices was caused by a combination of logistics (the study required signed patient consent); perceived concern about distress from the concept of an admission being 'avoidable' and patients whom the practice felt were too unwell in the aftermath of an admission to wish to be involved. There was variation in the quality of the competed SEA report, and as the SEA meetings were confidential to the practice and not observed, it was not possible to draw conclusions about the quality of the conduct of the SEA meetings within each practice. Practices in the study were representative of English practices in terms of size and markers of deprivation, although they were more likely to be rural. It is possible that practices recruited into this study may differ in other ways from those who declined to take part. The low proportion of patients consenting may also have introduced bias.

### Strengths and weaknesses in relation to other studies, discussing important differences in results

Most work on ACSCs uses panels of physicians examining diagnostic codes only, which risks leading to forming theories and conclusions without taking into account the individual factors surrounding the admission. Our research concurs with others using a similar approach of case-based analyses, which have reported lower rates of preventable admission compared with those predicted, by using ACSC lists.[27–29] Our findings question the validity of the current list of NHS ACSC codes, and others have questioned the assumption that unplanned admissions for ACSCs are reasonable indicators of performance.[34]

### Meaning of the study: possible explanations and implications for clinicians and policy-makers

The ACSC list used by the NHS is not reproducible in this small-scale study when unplanned admissions are examined in a detailed and systematic way by primary care teams caring for these patients. This may have international relevance, as similar lists of ACSCs are used in other countries. Preventable admissions identified in this study by primary care are less frequent than those determined from the current NHS list of ACSCs, although this was a relatively small study, and there is a need for larger scale study to confirm these findings. Lists of preventable admissions may more usefully classified by processes (such as drug error, failure of follow-up) than diagnostic codes which may be a blunt instrument. The NHS should consider not using the current list of ACSCs for performance management of unplanned admissions, at least until a more robust list has been developed and validated. Any new criteria should draw on data from actual cases and involve all relevant providers of healthcare and patients.

### Unanswered questions and future research

A larger study of similar design including the views of patients, secondary care and also the emergency services (when they were involved in admission) would be able to produce an updated and comprehensive list of preventable admissions and the processes involved, and also validate our findings. This would inform efforts to reduce unplanned admission by identifying context-specific factors related to admission, which current ACSC lists do not provide.

**Acknowledgements** We are very grateful to the Norfolk and Suffolk Primary andCommunity Care Research Office, the Sponsor NHS South Norfolk CCG, Public andPatient Involvement in Research (PPIRes) and the two patient representatives,the participating general practices, hospitals and patients for their help with theoriginal study.

**Contributors** RF: original idea for the study, supervised all aspects of the research, led the stakeholder liaison process, study design, data analysis and interpretation of results, and led the drafting of the paper. He is the guarantor. AHa: senior research associate for this study; responsible for the day-to-day running of the study, including steering committee, data collection and interpretation, and manuscript preparation. PKM, SP, AL, NS, AHo: study design, steering committee, data interpretation and manuscript preparation. GMP: lead statistician for the study; study design, steering committee, data analysis and interpretation, and manuscript preparation.

**Funding** This paper presents independent research funded by the National Institute for Health Research (NIHR) under its Research for Patient Benefit (RfPB) Programme (grant reference number PB-PG-0212-27059).

**Disclaimer** The views expressed are those of the author(s) and not necessarily those of the NHS, the NIHR or the Department of Health. The funders had no role in the study design and the collection, analysis, and interpretation of data and the writing of the article and the decision to submit it for publication. All researchers had full access to all of the data.

**Competing interests** All authors received a research grant from the National Institute for Health Research who funded this study and in addition to this: NS, GMP and RF report other grants from National Institute for Health Research, during the conduct of the study. AL, AHo, AHa, SP and PKM have nothing to disclose.

**Patient consent** Not required.

**Ethics approval** Ethics approval was granted by National Research Ethics Committee East of England, ref 13/EE/0328.

**Provenance and peer review** Not commissioned; externally peer reviewed.

**Data sharing statement** Data used in this research are presented in table 2.

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
