## [Reviewer comments · BMJ Open]

ARTICLE DETAILS

TITLE (PROVISIONAL)	Does practice analysis agree with the ambulatory care sensitive conditions list of avoidable unplanned admissions: cross-sectional study in the East of England
AUTHORS	Fleetcroft, Robert; Hardcastle, Antonia; Steel, Nicholas; Price, Gill; Purdy, Sarah; Lipp, Alistair; Myint, Phyo; Howe, Amanda

VERSION 1 – REVIEW

REVIEWER	Andrew Wilson University of Leicester UK
REVIEW RETURNED	20-Dec-2017

GENERAL COMMENTS	This paper reports a novel study on an important topic, which will be of interest to primary and secondary care practitioners as well as policy makers. It is well written and easy to follow. I have some suggestions for improvements, as below. Major points 1. To some extent the findings could have been predicted: it seems obvious that a detailed audit examining individual circumstances will produce different (and more accurate) findings on preventability than relying on diagnostic codes alone. What level of agreement would be 'fit for purpose'?2. The authors could make it clearer whether their aim was to improve the list of ACSC (ie by adding or dropping diagnoses) or to examine whether the concept itself is unhelpful. This is relevant for the authors' suggestion that findings should be confirmed in a larger study.3. If one accepts that SEA is closer to a gold standard than ACSC, would it be useful to summarise results in a 2x2 contingency table and present sensitivity/specificity/PPV/NPV of an ACSC diagnosis? I calculate these to be 65%, 77%, 31% and 93% respectively.4. Tables 2 and 3 are quite difficult to assimilate. Could some summary values be added to assist interpretation? Minor points 5. In the introduction and discussion it might be worth noting that others have distinguished between preventability (through preventive actions usually in primary care, eg maximising asthma control) and avoidability (through having alternatives to admission at the time of event, eg intermediate care), and that ACSC focus on the
---

	former. http://qualitysafety.bmj.com/content/early/2013/07/31/bmjqs-2013-002003.full 6. It is reported that 110 practices were approached and 20 recruited. It would be useful to know what sample size was originally planned, and perhaps to discuss the possibility of selection bias in the strengths and weaknesses section. 7. Although the low proportion of patients consenting and reasons for this are discussed under strengths and weaknesses, this section could also discuss the biases this may have introduced. 8. Is reference 5 correct for ACSC?
--	--

REVIEWER	Abdelouahab Bellou Beth Israel Deaconess Medical Center, Harvard Medical School
REVIEW RETURNED	26-Dec-2017

GENERAL COMMENTS	The authors report an interesting study to determine which emergency unplanned admissions are potentially avoidable and to compare to the NHS list of ambulatory care sensitive conditions (ACSC) managed in primary care doctors practices. They found a disagreement between the NIHS list and practice conclusions of potentially avoidable admissions. The major conclusion is that the ACSC conditions may not be valid. This study brings interesting finding but I found some weaknesses that authors should discuss. Here are my evaluation: 1- Title: no comment 2- Abstract: no comment 3- Methods: My major concern is that there is no external evaluators of the included cases which could have induced a bias. Authors should explain why they didn't use blinded external evaluators. This could explain the low number of cases included. The strenght of the study was the analysis of the review of the patients record rather than administrative and ICD codes. The use of Significant Event Audit method increases the value of the study. But it seems that a blinded methodology was not used. The authors should clarify. Results: the low number of included patients could have induced bias and limited the conclusions raised by the authors who mentioned this limitation in their manuscript. In table 3, authors should identify clearly the codes. For all tables, authors should define all abbreviations. Conclusions: no comment
--

VERSION 1 – AUTHOR RESPONSE

Dear Hemali Bedi,

Thank you for your invitation to revise this manuscript, and for the very helpful comments from the two peer reviewers. We have included a revised manuscript with track changes, and here are our responses to their questions:

Reviewer: 1

Reviewer Name: Andrew Wilson

This paper reports a novel study on an important topic, which will be of interest to primary and secondary care practitioners as well as policy makers. It is well written and easy to follow. I have some suggestions for improvements, as below.

Major points

1. To some extent the findings could have been predicted: it seems obvious that a detailed audit examining individual circumstances will produce different (and more accurate) findings on preventability than relying on diagnostic codes alone. What level of agreement would be 'fit for purpose'?

Response: We would have expected a moderate or substantial agreement between avoidability conclusion by practices using SEA and that according to the NHS ACSC code. In the interpretation of Kappa Viera et al [2005] state that moderate or substantive agreement would have a Kappa between 0.41 and 0.80, whereas we found a Kappa of only 0.253. We have included this paper by Viera et al as an additional reference number 33, and have modified the sentence in the results section on page 9 to read 'Guides to interpretation of values of Kappa suggest only 'fair' agreement between raters for 0.253, and we would expect a moderate or substantial agreement with a Kappa between 0.41 and 0.80.'

2. The authors could make it clearer whether their aim was to improve the list of ACSC (i.e. by adding or dropping diagnoses) or to examine whether the concept itself is unhelpful. This is relevant for the authors' suggestion that findings should be confirmed in a larger study.

Response: Our aim was '[to compare] the estimated avoidability of a list of unplanned admissions using two methods of assessment, those admissions deemed avoidable by the ACSC list and by a detailed notes review.' Now we have our results, we feel that the ACSC list may still be useful, but the current ACSC list needs revising in light of our preliminary findings and may need to reflect context specific factors. We have revised the section titled unanswered questions and future research to read 'A larger study of similar design including the views of patients, secondary care and also the emergency services (when they were involved in admission) would be able to produce an updated and comprehensive list of preventable admissions (ACSCs) and the processes involved, and also validate our findings.'

3. If one accepts that SEA is closer to a gold standard than ACSC, would it be useful to summarise results in a 2x2 contingency table and present sensitivity/specificity/PPV/NPV of an ACSC diagnosis? I calculate these to be 65%, 77%, 31% and 93% respectively.

Response: This is an interesting idea which we have considered carefully, however we have some reservations at the moment with assuming SEA is the 'gold standard'; If SEA is to be considered the gold standard, we feel that in addition to using the practice team in SEA, we would also want to include input from secondary care clinicians looking after the patient, the patient themselves, and also the emergency services. This statistical approach would be very useful for a subsequent larger study if indeed it included those additional components to the process of SEA. We have modified one of the sentences in the strengths and weakness of the study section to read 'Weaknesses include that the secondary care input to this process was limited to the discharge information received by primary care, and there was no input from the patient or where relevant the emergency services.'

We have also modified the sentence in the section on Unanswered questions and future research to read 'A larger study of similar design including the views of patients, secondary care and also the emergency services (when they were involved in admission) would be able to produce an updated and comprehensive list of preventable admissions and the processes involved, and also validate our findings.'

4. Tables 2 and 3 are quite difficult to assimilate. Could some summary values be added to assist interpretation?

Response; we have amended the headings of columns so they are consistent between both table 2 &

3. We have added summary statistics to the foot of table 2. We have added a description of disease

codes to table 3, and we have summarised the data in table 3 by ICD 10 disease codes. We have defined the abbreviations used at the foot of each table.

Minor points

5. In the introduction and discussion it might be worth noting that others have distinguished between preventability (through preventive actions usually in primary care, eg maximising asthma control) and avoidability (through having alternatives to admission at the time of event, eg intermediate care), and that ACSC focus on the former. <http://qualitysafety.bmj.com/content/early/2013/07/31/bmjqs-2013-002003.full>

Response: Thank you for this suggestion, we have including this reference in the introduction, and included the following sentence; 'Others have distinguished between avoidability (through having alternatives to admission at the time of event, eg intermediate care) and preventability (through preventive actions usually in primary care, eg maximising asthma control).'

6. It is reported that 110 practices were approached and 20 recruited. It would be useful to know what sample size was originally planned, and perhaps to discuss the possibility of selection bias in the strengths and weaknesses section.

Response: The study from which the data for SEAs was obtained examined the feasibility of using SEA in primary care to analyse emergency admissions, thus there was no sample size calculation. We have included the following information on the practices recruited to the study in the methods section;

'The practices included in the study were representative of English practices with respect to practice size and markers of deprivation, however they were more likely to be rural. The practices included had a mean of 8,640 registered adult patients (range 3,362 to 16148, national mean 8,316). 70% practices were classified 'urban', 20% practices 'town and fringe' and 10% were 'rural' (national averages 85%, 11% and 3% respectively). The mean Index of Multiple Deprivation score for practices was 21.1, (range 7.3 to 51.9, national mean 24.0, higher is more deprived).'

In the 'strengths and weaknesses' section we have also included the following: 'Practices in the study were representative of English practices in terms of size and markers of deprivation, although they were more likely to be rural. It is possible that practices recruited into this study may differ in other ways from those who declined to take part.'

7. Although the low proportion of patients consenting and reasons for this are discussed under strengths and weaknesses, this section could also discuss the biases this may have introduced.

Response: we have added the following sentence to the strength and weaknesses section; 'The low proportion of patients consenting may also have introduced bias.'

8. Is reference 5 correct for ACSC?

Response: Yes, this is described on page 163 of this reference, para 2 line 11. This is the earliest reference we are aware of to the use of ACSC.

Reviewer: 2

Reviewer Name: Abdelouahab Bellou

Institution and Country: Beth Israel Deaconess Medical Center, Harvard Medical School

Please state any competing interests: None

Please leave your comments for the authors below

The authors report an interesting study to determine which emergency unplanned admissions are potentially avoidable and to compare to the NHS list of ambulatory care sensitive conditions (ACSC) managed in primary care doctors practices. They found a disagreement between the NIHS list and

practice conclusions of potentially avoidable admissions. The major conclusion is that the ACSC conditions may not be valid.

This study brings interesting finding but I found some weaknesses that authors should discuss.

Here are my evaluation:

1- Title: no comment

2- Abstract: no comment

3- Methods:

My major concern is that there is no external evaluators of the included cases which could have induced a bias. Authors should explain why they didn't use blinded external evaluators. This could explain the low number of cases included. The strength of the study was the analysis of the review of the patients record rather than administrative and ICD codes. The use of Significant Event Audit method increases the value of the study. But it seems that a blinded methodology was not used. The authors should clarify.

Response: The cases of admission were randomly selected by the data clerks in secondary care to avoid selection bias by primary care. It is not possible to have external evaluators in SEA, this is because the SEA process needs to be conducted by the clinical team involved with the case [and external evaluators will not have been], and therefore this cannot be blinded [this is explained in more detail in reference number 30, Pringle M, Bowie P.]. It is possible that clinicians involved in SEA may be reluctant to disclose error, but we don't think this would introduce bias because if this happened it would be independent of the diagnosis.

We do agree that input from others involved in the case would increase the robustness of SEA, and we have acknowledged this in the strengths and weaknesses of the study paragraph, where we state 'Weaknesses include that the secondary care input to this process was limited to the discharge information received by primary care, and there was no input from the patient or where relevant the emergency services.'

We have amended the first sentence in strength and weaknesses of the study to read: 'The strength of the study was the analysis of the review of the patients record rather than administrative and ICD codes, and this analysis was performed soon after discharge by the primary care team looking after the patient, and as far as we are aware this is the first time this approach has been used in primary care.'

Results: the low number of included patients could have induced bias and limited the conclusions raised by the authors who mentioned this limitation in their manuscript.

Response: we agree that the relatively small number of patients could introduce bias, and that this is acknowledged in the Strengths and weaknesses of the study section

which include the following sentence; 'This was a relatively small sample of unplanned admissions, and a larger sample will have provided more information on a greater range of conditions, on the types of preventable admission and what factors may be involved.' We also refer to this in the paragraph on unanswered questions and future research

'A larger study of similar design including the views of patients, secondary care and also the emergency services (when they were involved in admission) would be able to produce an updated and comprehensive list of preventable admissions and the processes involved, and also validate our findings'

In table 3, authors should identify clearly the codes. For all tables, authors should define all abbreviations.

Response: we have added full diagnoses for each code. We have defined all abbreviations at the foot of each table.

Conclusions: no comment

VERSION 2 – REVIEW

REVIEWER	Andrew Wilson University of Leicester, UK
REVIEW RETURNED	07-Feb-2018

GENERAL COMMENTS	The authors have responded adequately to my comments,
---

REVIEWER	Abdelouahab Bellou Emergency Medicine Department, Beth Israel Deaconess Medical Center, Harvard Medical School, Boston, USA.
REVIEW RETURNED	11-Feb-2018

GENERAL COMMENTS	Authors addressed all concerns are raised in my first review. The draft was significantly improved. Despite a low number of cases analysed, information brought by the authors will be helpful for further studies that will confirm the results found in this study.
---